# Autism risk linked to prematurity is more accentuated in girls

**Leora Allen[1,2], Odelia Leon-Attia[1], Meirav Shaham[1], Shahar Shefer[1], Lidia V. Gabis**[1,3]*

**1** Weinberg Child Development Center at Safra Children's Hospital, Sheba Medical Center, Tel Hashomer, Ramat Gan, Israel, **2** Arrow Project, Sheba Medical Center, Tel Hashomer, Ramat Gan, Israel, **3** Sackler Faculty of Medicine at Tel Aviv University, Tel Aviv, Israel

* Lidia.Gabis@sheba.health.gov.il

## Abstract

### Introduction

Prematurity has been identified as a risk factor for Autism Spectrum Disorder (ASD). The link between Autism Spectrum Disorder (ASD) and birth-week has not been strongly evidenced. We evaluated the correlation between the degree of prematurity and the incidence of autism in a cohort of 871 children born prematurely and followed from birth. The cohort was reduced to 416 premature infants born between 2011–2017 who were followed for 2–14 years, and analyzed according to birth week (degree of prematurity), and according to gender.

### Results

43 children (10.3%) received a definite diagnosis of ASD. There was a significant correlation between birth week and the risk of ASD, with 22.6% of children diagnosed with ASD when born at 25 weeks, versus 6% of ASD diagnoses at 31 weeks of prematurity. For children born after 32 weeks, the incidence decreased to 8–12.5%. A strong link was found between earlier birth week and increased autism risk; the risk remained elevated during near-term prematurity in boys. A correlation between early birth week and an elevated risk for ASD was seen in all children, but accentuated in females, gradually decreasing as birth week progresses; in males the risk for ASD remains elevated for any birth week.

### Conclusion

A statistically significant increase in rates of autism was found with each additional week of prematurity. Females drove this direct risk related to degree of prematurity, while males had an elevated risk throughout prematurity weeks, even at near-term. We recommend including ASD screening in follow up of infants born prematurely, at all levels of prematurity.

**Data Availability Statement:** All relevant data is available within the paper and its Supporting Information files.

**Funding:** The author(s) received no specific funding for this work.

**Competing interests:** The authors have declared that no competing interests exist.

**Abbreviations:** ADHD, Attention Deficit/Hyperactivity Disorder; ADOS, Autism Diagnostic Observation Schedule; ASD, Autism Spectrum Disorder; BSID-II, Bayley Scale of Infant Development- second edition; CAT/CLAMS, Clinical Adaptive Test/Clinical Linguistic and Auditory Milestone Scale; CP, Cerebral Palsy; DDST-II, Denver Developmental Screening Test, second edition; DQ, Developmental Quotient; GDD, Global Developmental Delay; K-BIT, Kaufman Brief Intelligence Test; LIPS-R, Leiter International Performance Scale—Revised; MSEL, Mullen Scales for Early Learning; SB-4, Stanford-Binet Fourth Edition; WISC-R, Wechsler Intelligence Scale for Children–Revised; WPPSI-III, Wechsler Preschool and Primary Scale for Intelligence–Third Edition.

## Introduction

The prevalence of Autism Spectrum Disorder (ASD) has been increasing [1], up to 1:54 in recent CDC reports. It is a disorder which affects 1 in 42 males and 1 in 189 females [2]. ASD is a neurodevelopmental disorder that is characterized by deficits in social behaviors and communication skills, as well as the presence of certain stereotyped or repetitive behaviors and/or interests [1]. A definitive diagnosis of ASD is typically made by the age of two to three, although signs of the disorder may be seen earlier [3]. ASD is caused by a variety of both genetic and environmental factors [4, 5] and it is associated with brain insult and inflammation [6]. Prematurity, defined as a baby born before the 37th week of gestation, has been identified as one environmental risk factor for ASD, either as a primary risk factor or an additional risk in families with a sibling with ASD [7]. Even though the call for action in the publication "*Born Too Soon*: *The Global Action Report on Preterm Birth"* in 2012 reported that in 2010, approximately 1 in 10 babies was born prematurely, the incidence of prematurity did not change significantly in Israel (7.8–7.4% in 2015) [8], and continued to rise in USA with 11.5% reported in 2012 [9].

Thus, this population of prematurely born children warrants further investigation due to their significant risk of developmental delay. Common consequences of prematurity include medical complications shortly after birth [10] as well as neuromuscular disorders such as cerebral palsy, intellectual disability, and respiratory conditions [9]. It has been shown that routine neonatal follow up is the best way to monitor neurodevelopmental progression in babies born prematurely [11]. Certain factors, such as male gender, cerebral palsy, and low birth weight, have been shown to correlate with poor neurodevelopmental outcome in premature infants [12].

ASD has been known as a disability with a gender skew, with males predisposed to receive a diagnosis of ASD up to three to four times more than females [13]. This could be due to an under-diagnosis of females with autism [14], or to a true male gender genetic tendency and vulnerability. Comorbidities including other neurologic disorders and developmental disorders are common [15], such as Cerebral Palsy (CP) and global developmental disability (GDD) [16]. Studies have shown that there is a higher incidence of ASD in patients with non-spastic CP [17] and the comorbidity of ASD and GDD is well established.

It has been shown that when two conditions appear together, the symptoms are more severe [18]. The link between earlier gestational week and increased risk of autism was proven in prior studies, however the correlation was defined in three prematurity defined categories (24–26 weeks–extremely low gestational age infants; 27–33 weeks–preterm infants; 34–36 weeks–late preterm infants) as compared to two term categories (37–41 weeks–term infants; 42 weeks–post-term infants) and with each earlier group, the risk was higher. However, the groups were not clumped per weeks but per number of children and the results represent risk per category (two weeks between 24–26 weeks and for six weeks between 27–33 weeks) and without gender differences [19].

Other studies suggesting a link between prematurity and autism use birth weight as proxy, [20, 21] however, birth week does not correctly represent prematurity level as related to brain maturity, and even birth weight at term was found to be related to autism risk [22].

In this study, we aimed to identify whether there is a correlation between the birth week and a diagnosis of ASD, and if there is a gender predisposition in this correlation. Additionally, we compared it to two common diagnoses in prematurity—of CP and GDD, in a cohort of children born prematurely in Israel, and followed at one tertiary referral center.

## Materials and methods

This study was performed at the prematurity clinic of the Weinberg Child Development Center, Edmond and Lilly Safra Children's Hospital, Tel Hashomer, which is the largest referral center in Israel and linked to a tertiary prematurity unit.

Infants born below 36 weeks are referred for neurodevelopmental follow-up at 4–8 weeks after discharge. As such, the cohort is unbiased by the subsequent diagnoses. Patient data was recorded following patient visits. Data from 2011–2017 was examined retrospectively for all infants at least two years of age at the completion of the study, and their developmental follow-up assessments were documented. After the cohort of babies born prematurely was established, diagnoses were identified. Though a definitive diagnosis of ASD is given around the age of two to three years, patients who have normal development through the age of two years are oftentimes discharged from care at the development centers. Therefore, we only included patients who were examined until at least the age of two years in order to account for some potential loss due to follow up. Of interest in this cohort were diagnoses of ASD, as compared to diagnoses of Global Developmental Delay (GDD), and Cerebral Palsy (CP). A diagnosis of ASD was given based on DSM-IV-TR or DSM-5 criteria [23, 24]. The Autism Diagnostic Observation Schedule—ADOS [25] was used to confirm diagnoses. Children also underwent a neurodevelopmental evaluation [26]. A Developmental Quotient (DQ) score was obtained for each child based on the Denver Developmental Screening Test, second edition (DDST II); [27] and the Clinical Adaptive Test/Clinical Linguistic and Auditory Milestone Scale (CAT/CLAMS) [28]. Other diagnoses such as global developmental delay and cerebral palsy were made according to standardized diagnostic tools including neurodevelopmental examination and neuropsychological evaluation. Specifically, a diagnosis of GDD was based on delays by more than 2 SD in two or more developmental areas (DQ < 70), both on neurodevelopmental examination and neuropsychological evaluation [29]. A diagnosis of CP was made by the Pediatric Neurologist based on neurological examination, imaging and Gross Motor Function Classification System, according to diagnostic guidelines [30].

All participants underwent standardized cognitive testing. Specific instruments selected for cognitive testing were dependent on the child's age and functioning level. Instruments used included the Bayley Scale of Infant Development- second edition (BSID-II) [31], Mullen Scales for Early Learning (MSEL) [32], or a cognitive test such as the Stanford-Binet Fourth Edition (SB-4) [33], Wechsler Preschool and Primary Scale for Intelligence–Third Edition (WPPSI-III) [34], Wechsler Intelligence Scale for Children–Revised (WISC-R) [35], or Kaufman Brief Intelligence Test (K-BIT) [36]. Diagnoses were analyzed according to birth week and gender.

### Ethical approval

The study has been approved by the appropriate institutional research ethics committee "Sheba Medical Center" Institution Review Board- Helsinki approval number is 3592-16-SMC. For this type (retrospective) of study, formal consent is not required.

### Statistical analysis

In order to examine the association between birth week, gender and diagnoses, we used a Wilcoxon test for multiple comparisons assuming non parametric distributions. In addition, a nominal logistic model was fit using the following factors: gender, birth week and birth week X gender interaction. In order to examine the relationship between birth week and birth weight we fit a linear regression model. All statistical analyses were conducted using SPSS version 21 and JMP Pro14 By SAS.

## Results

Of the 838 children identified who were born premature, 416 children were seen after the age of two years; 422 children were excluded. Of the excluded 229 (54.3%) were male, and 193 (45.7%) were female. The average birth week was 32.3 weeks (SD = 2.5) and the average birth weight was 1670 grams (SD = 486), which is significantly higher compared to children who were seen after the age of two years, t(834) = 7.26, p<0.001, t(819) = 6.65, p<0.001, respectively.

Of the 416 children seen after the age of two years, 246 (59.1%) were male, and 170 (40.9%) were female. 177 (42.5%) of the premature children were twins (4 of them identical), and seven (1.7%) from triplet pregnancies. With an age range from 2–14.2 years old, the average age was 4.2 years (SD = 2). Birth week ranged from week 24–37, with an average of 30.8 weeks (SD = 3.3). Birth weight ranged from 368–3550 grams, with an average of 1427 grams (SD = 557). The relationship between birth week and birth weight exhibited a significant (*p<0.001*) linear correlation for both males and females using a linear regression model. Of the 416 children, 43 (10.3%) received a diagnosis of ASD, 62 (14.9%) received a diagnosis of CP and 68 (16.3%) received a diagnosis of GDD. Additional diagnoses such as ADHD are usually decided after the age of four years and more significantly at school-age. Accordingly, our cohort is limited and we will report only on the three major diagnoses–ASD, CP and GDD.

There was a significant difference in the probability of ASD diagnosis, CP and GDD between birth weeks $\chi^2$ (13) = 22.5, p < .05, $\chi^2$ (13) = 86.7, p < .001, and $\chi^2$ (13) = 26.5, p < .05, respectively. The rates of ASD and GDD had higher percentage of babies born no later than week 29 diagnosed with these disorders. The rates of CP are increased with earlier prematurity and decreased by weeks 32 (See Fig 1).

The primary outcome in this study was a diagnosis of ASD. Of the patients with ASD, 27 (62.8%) were males and 16 (37.2%) were females. The average age of the patients in the cohort was 5.7 years old (SD = 2.2). The age of initial ASD diagnosis was in the range of 1.1–8.5 years, with mean age of 3.5 years (SD = 1.8). Of the 43 children with ASD, 19 (44.2%) were twins. The average birth week was 30.2 weeks (SD = 3.7) and the average birth weight of the patients with ASD was 1415 grams (SD = 704). When tested for both male and female, it appears that the probability for ASD increases with earlier birth weeks: There is an increased probability of

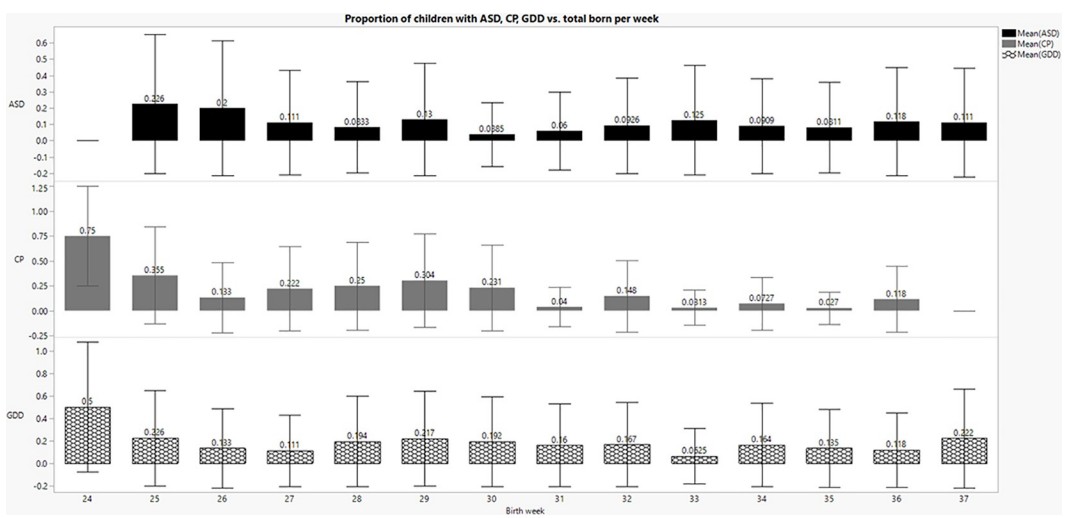

**Fig 1. Rates of diagnosis of ASD, GDD, and CP by specific birth week.**

ASD with each reduction in the week of prematurity between the weeks of 25 to 31, with the probability of ASD at 22.6% at 25 weeks versus 6% at 31 weeks, *Wilcoxon approximately Z = 2.1, p<0. 05*. Moreover, early ranges of birth weeks are significantly different from later ones. Weeks 25–26 have a higher probability of ASD– 20%-22.6%, as compared to weeks 27–37, with a probability of 8.1%-13% (except week 30 with 3.8% and week 31 with 6%). (See Table 1 and Fig 1).

We examined the extent to which gender contributed to the findings. There was a statistically significant difference between birth week of females born prematurely with ASD (M = 27.9, SD = 3.4 weeks) versus females born prematurely who did not receive a diagnosis of ASD (M = 30.9, SD = 3.2 weeks), *t(168) = 3.60, p < .001*. On the contrary, there was no significant difference between birth week in males born prematurely with ASD (M = 31.7, SD = 3.2 weeks) as compared to males born prematurely who did not receive a diagnosis of ASD (M = 31.1, SD = 3.3 weeks), *t(244) = 0.99, p>.05*.

In order to compare multiple pairs of ASD versus birth week, we performed a multiple comparisons Wilcoxon test. When gender was taken into account, it is clear that the significance for early birth week and higher probability of ASD is observed in females. There is a significant increase in rates of ASD in earlier weeks for females; $\chi^2$ *(13) = 44.6, p < .001*. However, for males there was an elevated probability, between 6.7% to 23.5%, regardless of level of prematurity, with no significance $\chi^2$ *(13) = 6.6, p>.05* (See Table 1 and Fig 2). For females there is increased probability with each reduced week of prematurity between weeks 25 and 32, with the probability of ASD at 31.3% in 25 week versus 0% at 32 week, *Wilcoxon approximately Z = 2.7, p<0. 01*. Weeks 25–27 have a significantly higher probability of ASD– 25%-31.3%, as compared to weeks 31–35, with a probability of 0%-5%, p value range between 0.001 to 0.03.

The proportion of children with ASD born below 33 weeks was 41% of all ASD for boys and 32% for girls, almost 1:1 ratio, however- after 33 weeks, 20% for males and 4% for females, reminding the known 1:5 male/ female ratio of idiopathic autism [37] (See Table 1).

**Table 1. Number and percentage of children receiving a diagnosis of ASD by gender based on birth week.**

| Birth week (n:M,F) | ASD MALE | ASD FEMALE | TOTAL ASD |
|---|---|---|---|
| | (n = 27) | (n = 16) | (n = 43) |
| | N(%) | N(%) | |
| 24 (n = 4:2,2) | 0 (0%) | 0 (0%) | 0 (0%) |
| 25 (n = 31:15,16) | 2 (13.3%) | 5 (31.3%) | 7 (22.6%) |
| 26 (n = 15:7,8) | 1 (14.3%) | 2 (25%) | 3 (20%) |
| 27 (n = 27:16,11) | 0 (0%) | 3 (27.3%) | 3 (11.1%) |
| 28 (n = 36:24,12) | 2 (8.3%) | 1 (8.3%) | 3 (8.3%) |
| 29 (n = 23:10,13) | 2 (20%) | 1 (7.7%) | 3 (13%) |
| 30 (n = 26:16,10) | 0 (0%) | 1 (10%) | 1 (3.8%) |
| 31 (n = 50:30,20) | 2 (6.7%) | 1 (5%) | 3 (6%) |
| 32 (n = 54:34,20) | 5 (14.7%) | 0 (0%) | 5 (9.3%) |
| 33 (n = 32:17,15) | 4 (23.5%) | 0 (0%) | 4 (12.5%) |
| 34 (n = 55:35,20) | 4 (11.4%) | 1 (5%) | 5 (9.1%) |
| 35 (n = 37:24,13) | 3 (12.5%) | 0 (0%) | 3 (8.1%) |
| 36 (n = 17:9,8) | 1 (11.1%) | 1 (12.5%) | 2 (11.8%) |
| 37 (n = 9:7,2) | 1 (14.3%) | 0 (0%) | 1 (11.1%) |
| TOTAL (n = 416:246,170) | 27 (10.9%) | 16 (9.4%) | 43 (10.3%) |

*The percentages in parentheses represent rations of same week and gender, and thus do not add up to total ASD count.

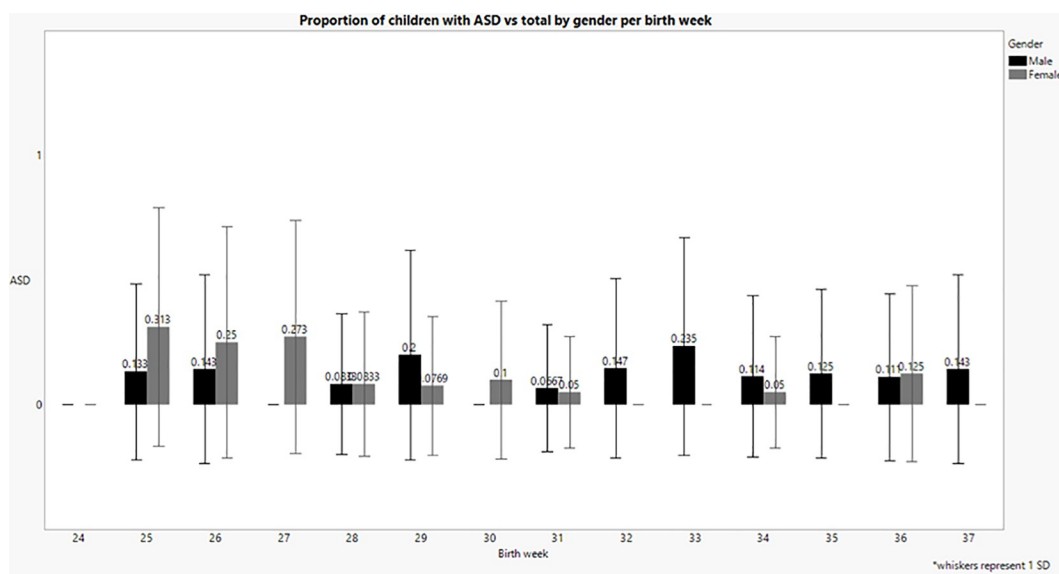

**Fig 2. Differences in rates of ASD between males and females by birth week.**

On top of the multiple non parametric comparisons for each pair using the Wilcoxon method, we constructed a nominal logistic model with the following factors: gender, birth week and the interaction of those effects. The model was significant $\chi^2(3) = 13.30$, $p<0.01$ as well as the main effect for birth week $\chi^2(1) = 4.61$, $p<0.05$, and the interaction effect, $\chi^2(1) = 9.53$, $p<0.01$. The main effect for gender was not significant $\chi^2(1) = 2.18$, $p>0.05$. For females we can see that as the week birth increases, the proportion for ASD reduces. For males, the picture is different: the proportion of ASD remains relatively constant. However, in earlier birth weeks, the risk for ASD is higher for females than males. As we move towards birth week 28 the proportion evens out, with the rate becoming slightly higher for males in the last increment (See Fig 3).

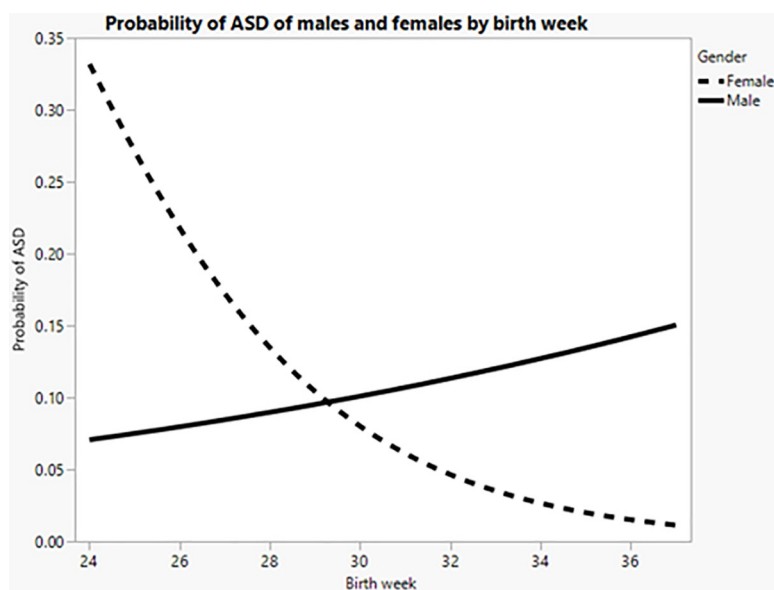

**Fig 3. Probability of ASD of males and females by birth week.**

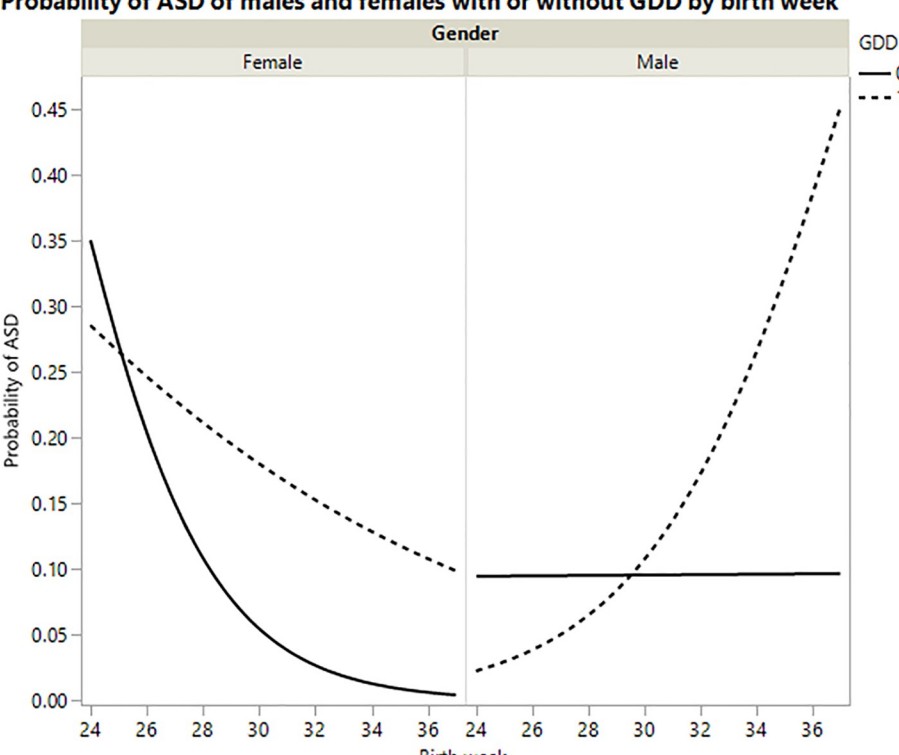

**Fig 4. Probability of ASD of males and females with or without GDD by birth week.**

68 children (16.3%) received a diagnosis of GDD, thirteen of them have ASD (19%). Adding the GDD as an intervening factor to the model, strengthened the model slightly $\chi^2(6) = 22.81, p<0.001$ and the GDD is significant $\chi^2(1) = 6.48, p<0.05$ as well as its interaction with birth week and gender $\chi^2(1) = 4.22, p<0.05$. When constructing a graph based on the nominal logistic model prediction, we see that while for females trends remain similar, the proportion of ASD drops as birth week increases. For females with GDD, the same trend is kept but with a higher propensity towards ASD. For males however, trends reverse. While for males without GDD the trend is relatively constant across birth weeks, for males with GDD the proportion of males with ASD increases with birth week (See Fig 4).

Additionally we examined the birth weight. There was a statistically significant difference between birth weight of girls born prematurely with ASD (M = 1009, SD = 532) as compared to girls born prematurely who did not receive a diagnosis of ASD (M = 1350, SD = 504), $t(167) = 3.56, p < .05$. On the contrary, there was no significant difference between birth weight of males born prematurely with ASD (M = 1656, SD = 690 weeks) when compared to males born prematurely who did not receive a diagnosis of ASD (M = 1482, SD = 556), $t(243) = 1.48, p>.05$.

We also examined IUGR/ SGA. Thirty seven children had IUGR/ SGA, 20 were males (54%) and 17 were females (46%). Birth week ranged from week 24–37, with an average of 30.8 weeks (SD = 3.9). Birth weight ranged from 580–2200 grams, with an average of 1130 grams (SD = 448).

No correlation was found between IUGR/SGA and ASD, $\chi^2 (1) = 0.2, p>.05$., Thirty four of IUGR without ASD (92%) and 3 IUGR with ASD (8%). In order to examine whether IUGR

had an effect on the results we exclude these 37 children from model. The model was significant $\chi^2(5) = 17.74$, $p<0.01$, effect for IUGR was not significant $\chi^2(1) = 0$, $p>0.05$ and the conclusions remained the same: the probability to ASD decreased for females with each birth week progress while for males it remained relatively constant.

We also identified whether the births were multiples or singletons. For the purpose of the statistical analysis, the twins and triplets were combined into a single group. Of all the premature infants in the cohort, 31% of twins were born in weeks 32–33, 24% born in weeks 30–31, and 22.8% born in weeks 34–35. Compared with the singleton births, there was a significant difference in the birth week, $t(411) = 3.9$, $p< 0.001$. Mean birth week of singletons was 31.5 (SD = 2.7) versus 30.3 weeks (SD = 3.6) in multiple pregnancies. Out of 184 multiple pregnancies 19 had ASD (10.3%). We examined the distribution of the proportion of multiple pregnancies with ASD across birth weeks, but no particular significance was found between earlier and later birth weeks, nor were we able to construct a significant model indicating twins as a significant intervening factor $\chi^2 (1) = 1.52$, $p>.05$.

There was no significant difference found between birth weight of singletons when compared to multiple pregnancies, $t(411) = 1.5$, $p> 0.05$.

## Discussion

In this large scale study examining the correlation between autism and prematurity in Israel, we found a much higher prevalence—10.3% of ASD in the premature population compared to the known general incidence of 1.2–2%. This demonstrates that prematurity is a clear risk factor for the development of ASD. A large-scale study undertaken in Sweden in 2011 found a significant association between autism and prematurity, however they focused on the entire category of prematurity (i.e. born before 37th week of gestation) [38]. In our study, we separated the children by week of birth and by gender, revealing a significant increase in the rates of ASD with each week that the child was born earlier, especially before 31 weeks. The degree of prematurity was influenced more strongly by female gender, to the degree that the male to female ASD ratio is almost 1:1. However, after 33 weeks the incidence remains elevated and does not continue to decrease, following the pattern of increased risk in males, in the same gender distribution as "idiopathic" autism [39]. As demonstrated in Chart 1, there was a significant correlation between the birth week and diagnosis of ASD only during extreme prematurity. The main contribution to the decreasing slope is driven by female gender, thus males are at an elevated risk regardless of the level of prematurity, which is what has been seen in the literature until now [40]. This change in the slope tendency and the gender differences elute to a "double hit" theory in regards to prematurity and autism risk, with a higher vulnerability in premature males, possibly related to a genetic predisposition. In other words, girls are mainly at risk because of being born premature and boys are more predisposed by their gender, and are at a significantly elevated risk even at near-term. Differential brain vulnerability to prematurity based on sex differences was shown in other studies, and there is a clear differential gender vulnerability to complications of prematurity [41] such as retinopathy [42].

Brain gender differences in autism have been revealed as well. Sex differences in ASD are associated with cortical regions involved in language and social function [43]. Based on gender influence on prematurity complications and female and male differential brain development in autism, we may presume that prematurity complications may have a different influence on female preterm extrauterine brain maturation and development. This assumption may allude to the differential need for neuroprotection measures in boys and in girls born prematurely.

A limitation of our study is that we are a referral center, and therefore we lost some patients to follow up, and there are differences by birth weight and week between the group, due to the

fact that parents to premature infants at less risk, tend so arrive less to follow ups. Thus, it is possible that some of these children who were born prematurely were followed by primary care centers, especially for the less concerning group of patients born near-term (36–37 weeks) and with higher birth weight. An additional limitation is that almost half of the cohort were twins. Although analyzed separately and born significantly at earlier weeks than singletons, it is possible that some of the twins were monozygotic and as such have an additional reason for increased ASD risk [44, 45].

A third limitation is the lack of knowledge of the true incidence of ASD in Israeli children born at term. For this group there is an intrinsic referral bias to our center, and therefore it would be important in future studies to concurrently follow a term group from birth as we followed the premature infants. Our reference to compare the incidence of ASD in Israeli population is not recent enough, and does not reflect the increase in incidence published by the CDC for the study period [2, 46].

What this study demonstrates is that there is indeed a correlation between prematurity and autism, at any degree of prematurity. Early intervention has been shown to have significant impact on the prognosis and functioning of children with autism. Specifically, the earlier intervention is begun, the more impactful it is on the functioning of the child. It is important to be aware of prematurity as a significant risk factor for the development of ASD, and include ASD screening in each follow up of those infants born prematurely, until at least three years of age.

Because of the large size of our cohort, we were also able to look at the differences between singleton births and multiple births. We found that there was a significant difference in the birth week between singletons and multiples, specifically that multiple births occurred earlier. While this is not necessarily new information, it is indicative of the need to improve maternal health as it relates to multiple birth pregnancies. Additionally, since we have shown that prematurity is a risk factor for ASD, and since we know that multiple births carry a higher risk of prematurity [3], there is a greater need to monitor twins for longer periods of time to assure appropriate development.

Future areas of research will need to focus on eliminating some of the variables that might have affected our study. For instance, by following babies prospectively from birth, it is hopeful to eliminate some of the loss due to follow up. Additionally, the etiology of ASD is multifactorial. It will be important to compare key characteristics that can help with screening in children with ASD who were born premature versus children with ASD who were born at term and observe long term outcome as well. For example, one prospective study looked at very low birth weight ($<$ 1500 g) infants born prematurely, and found that according to the M-CHAT test, 26% of them were likely to develop autism [47]. This is a significant finding, and by undertaking a study with a larger cohort (similar to the current research), we hope to strengthen this claim.

Additionally, one area of research that is currently of utmost importance in the field of autism is understanding the relationship between genetics and the development of the disorder, with the possibility of prematurity as a 'double hit' when there is a genetic predisposition. By examining this population of children with ASD who were born prematurely, looking at other familial risk factors such as siblings with ASD who were not born prematurely, we might be able further elucidate the nature versus nurture influences. It has been shown that there is a genetic risk of autism [48], and our study may provide additional insight into gender specific development of ASD. This research supports the existent recommendations for early intervention, especially for those children born prematurely [49,50], hopefully leading to better outcomes.

## Supporting information

**S1 Data.**
(XLSX)

## Author Contributions

**Conceptualization:** Lidia V. Gabis.

**Data curation:** Leora Allen, Odelia Leon-Attia.

**Formal analysis:** Odelia Leon-Attia, Meirav Shaham.

**Methodology:** Odelia Leon-Attia, Lidia V. Gabis.

**Resources:** Shahar Shefer.

**Supervision:** Shahar Shefer, Lidia V. Gabis.

**Writing – original draft:** Leora Allen, Meirav Shaham, Lidia V. Gabis.

**Writing – review & editing:** Leora Allen, Odelia Leon-Attia, Meirav Shaham, Shahar Shefer, Lidia V. Gabis.

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
