## [Decision Letter · Decision Letter 0]

8 Apr 2020

PONE-D-19-35927

Autism risk linked to prematurity is more accentuated in girls

PLOS ONE

Dear Dr. Gabis,

Thank you for submitting your manuscript to PLOS ONE. As you can see, I have received three reviews from experts in the field, and they expressed a range of opinions about your manuscript. This is a critical area of research, and I commend your efforts. However, after careful consideration, we feel that it has merit but does not fully meet PLOS ONE’s publication criteria as it currently stands. Therefore, we invite you to submit a revised version of the manuscript that addresses the points raised during the review process.

I will not restate all reviewer comments however, please pay careful attention to two areas.  First, you will need to reconcile your work with the previous literature noted by reviewer 1. Please frame your work in this context, and differentiate your work from prior studies to highlight the contribution your work makes to the literature. Second, there were a number of analytic considerations noted. Please address each of these, either by including their suggested changes, or providing justification for not including them. 

We would appreciate receiving your revised manuscript by May 23 2020 11:59PM. To enhance the reproducibility of your results, we recommend that if applicable you deposit your laboratory protocols in protocols.io, where a protocol can be assigned its own identifier (DOI) such that it can be cited independently in the future. For instructions see: http://journals.plos.org/plosone/s/submission-guidelines#loc-laboratory-protocols

We look forward to receiving your revised manuscript.

Kind regards,

Eric J. Moody, Ph.D.

Academic Editor

PLOS ONE

Reviewers' comments:

Reviewer's Responses to Questions

**Comments to the Author**

1. Is the manuscript technically sound, and do the data support the conclusions?

Reviewer #1: No

Reviewer #2: Yes

Reviewer #3: Yes

2. Has the statistical analysis been performed appropriately and rigorously? 

Reviewer #1: No

Reviewer #2: Yes

Reviewer #3: Yes

3. Have the authors made all data underlying the findings in their manuscript fully available?

Reviewer #1: No

Reviewer #2: Yes

Reviewer #3: No

4. Is the manuscript presented in an intelligible fashion and written in standard English?

Reviewer #1: Yes

Reviewer #2: Yes

Reviewer #3: Yes

5. Review Comments to the Author

Reviewer #1: This study examines the association of ASD with preterm birth in 416 children from the ages of 2 to 11 years. More specifically, the authors seek to investigate the possibility of a gestational age gradient such that the risk of ASD decreases with each week of gestation from 24 to 37 weeks (from the lower boundary of extremely preterm to the upper boundary of later preterm) and how such a gradient might differ between males and females. They find that females have a significantly increased risk of ASD relative to males at lower gestational ages, whereas males maintain the same risk of ASD regardless of degree of prematurity.

Comments

1. The authors’ claim that such a gestational age gradient “has not been evidenced” in prior studies is inaccurate. At least two prior studies have demonstrated such a gradient (by week) with much larger epidemiological samples and with adjustment for a number of critical covariates and confounding variables that were not considered in the current study, presumably because the small sample size precluded their inclusion in statistical analyses. Please see Kuzniewicz et al., 2014 and Xie et al. (2017). In addition, Talmi et al. (2020) recently conducted a retrospective national cohort study of all children born in Israel from 2000-2012 using gestational age categories of extreme, very, moderate, and late preterm, showing a clear-cut gestational age gradient, as have a number of other large European population studies over the past decade, none of which are cited by the authors.

2. The authors do not adequately address the loss to follow up of more than half of the sample (n=871) that was referred to them and the potential bias that this introduces into their results. At the very least, demographic and birth characteristics between those retained and those lost should be compared.

3. Fetal growth restriction has been found to contribute, independently of gestational age, to ASD risk in a number of high-quality studies. The lack of consideration of FGR/SGA in this study is problematic, especially in that FGR may represent a distinct preterm phenotype that confers ASD risk by different mechanisms than GA.

4. The large age range is somewhat problematic in that the preterm ASD phenotype is not well characterized. ASD classifications in the youngest children, whether positive or negative, may not be stable.

5. It is now standard in autism epidemiological research to investigate ASD with and without intellectual disability (ID), as these are likely to differ to in etiology. The small sample size of the current study may have precluded meaningful statistical analysis addressing ID, but these data could have been presented for descriptive purposes. For example, one wonders how many of the EPT/VPT girls with ASD had ID and severe ID. The authors list a large number of cognitive measures, but they do not use these to describe their sample.

6. The authors state that all ASD diagnoses were confirmed with the ADOS, but do not report any ADOS data, for example, the Modules used – which would have given a better sense of the level of function in the sample – or the severity scores for repetitive behaviors as well as social communicative impairment. These data are of interest in the characterization of any preterm ASD sample, given that the ADOS, a gold-standard measure, has rarely been used to characterize ASD in premies.

7. The rationale for investigating the GA gradient for global development delay (GDD) and CP in addition to ASD is not clearly or compellingly explained. It is not clear if these are overlapping or mutually exclusive groupings, although the latter seems to be the case. In addition, it is not reported if any of the children classified as ASD had severe ID or severe gross motor impairment. This is problematic in that DSM-5 excludes an ASD diagnosis that is “better explained by intellectual disability (intellectual developmental disorder) or global developmental delay”, and it is not uncommon for EPT-born children to have this severity of ID. In addition, severe motor impairment (e.g., a Gross Motor Function Classification of Score = 5) could result in autism-like symptoms because of an inability to orient, direct attention, show, use gestures, etc.

8. The authors do not adequately account for multiple birth in their statistical analyses. Approximately, 44% of the sample and 44% of those diagnosed with ASD were multiples, with the large majorities of those twins. (Information on monozygosity was not available.) T-tests and chi-squares are used to analyze differences between singletons and multiples in mean GA and distributions by gestational age, respectively, but the results are difficult to follow and the authors do not offer a clear interpretation. It would seem more effective to have included this variable as a covariate in the logistic regression analysis. I also wondered how many of those classified with ASD were from the same birth.

9. It seems necessary to consider that the present findings are unreliable, given the small sample size and the inconsistency of the findings from a number of prior studies, which are not addressed in this report. It is notable, however, that others (e.g., Schendel & Bhasin; Xie et al., 2014) have found that females are at relatively higher risk of ASD with ID vs. without ID than males, which may bear on the current findings, if ID were taken into account.

Reviewer #2: This manuscript explores the relationship of birth week among premature infants to Autism Spectrum Disorder (ASD), to see if the degree of prematurity might predict the likelihood that ASD might develop. 416 children born from 2011 to 2017, from an original cohort of 871 premature children, followed for at least two years were studied retrospectively, looking at birth week and gender. All children were from a single child development referral center in Israel. Nearly half were twins or triplets, some of whom may have been monozygotic. The authors acknowledge that this may have affected the risk for developing ASD. A total of 43 children were felt to have ASD, and the risk was greatest for those who were more premature. While the overall risk of developing ASD is greater for males than females, an elevated risk of ASD with lower birth week was greatest for females, who had a higher risk than males for lower birth weeks and a lower risk than males for higher birth weeks. For males, the risk was relatively stable across the range of birth weeks.

ASD is felt to be due to a combination of genetic and environmental factors, with brain insult and inflammation possibly contributing to the risk of developing it. Factors related to premature birth may contribute to the environmental components contributing to the development of ASD. Lower birth weight correlated with ASD risk in females, but not in males. A possible correlation between Cerebral Palsy (CP) and Global Developmental Delay (GDD) was also looked for. Measurements of ASD, CP and GDD by gender are shown across the range of birth weeks in the Figures 1-3. Table 1 summarizes the incidence of ASD by gender and birth week. A number of measures of cognitive function were utilized. Statistical analysis was appropriate.

This study shows that prematurity is a risk factor for the development of ASD, and provides additional information regarding the relative risk attributable to birth week and gender. It is beyond the scope of this study to look at specific factors that might contribute to the risk of ASD in premature children, such as anoxic brain damage, intraventricular hemorrhage, presence or absence of Retinopathy of Prematurity (ROP), and treatment of Type 1 ROP with anti-VEGF injections. What can be said is that earlier intervention for those developing ASD may contribute to better outcomes, and recognition of the increased risk of ASD in premature children is therefore important. It is an interesting study.

Reviewer #3: This is an interesting study with a retrospective look at prematurity and gender risk for ASD. There are several things that can strengthen it to make it more meaningful to autism researchers, clinician, and parents.

Abstract: Children were followed for only “at least” two years, so the statement about girls should may need to be tempered to say that this relationship is evident only for girls whose symptoms are sufficiently apparent to be diagnosed by the age of 24 months, if that was the case (no info is given on when they were diagnosed). Girls with higher cognitive and language abilities are often not diagnosed until much later.

Introduction: When the manuscript is revised, the CDC prevalence data needs to be updated to 1:54. “significant risk of developmental risk” may need to be revised – is developmental delay the intended meaning?

Please clarify when the children were evaluated. It is stated that you only included children seen in the clinic up to the age of 2 to prevent loss to follow up, but several of the cognitive measures listed are for older children only. At what ages where the cognitive evaluations completed? It is apparent from the age range (2-14.2 years) that cognitive testing may have occurred at any point. This is a retrospective study, so presumably cognitive testing was done for a reason unrelated to the study, which should be addressed at least briefly to determine possibility of bias in the sample. For example, if the cognitive testing was performed as a matter of routine follow up for a premature child, that is less biased than if they were only performed when disability was suspected, introducing at least some selection bias in the sample. This issue is also related to the comment above about females across the spectrum who are typically not diagnosed until later in life.

How old were the children when they were diagnosed? That is an important proxy for interpretation of the severity of the ASD.

Also, it would be helpful to know co-morbid diagnoses. Co-morbidity data would also aid in the interpretation of the findings. Are the females diagnosed with ASD in the lower gestational ages also affected by multiple disabilities or is ASD the primary risk? All of this information is more helpful than a blanket statement about risk and females as if they are all equivalent in terms of ASD. The current data are helpful, but MORE data from the records would significantly improve the interpretation and application of the study. Since Cognitive testing was completed, that is one example of data that aids in interpretation. Females with Intellectual Disability (ID) present very differently than females without ID, and may or may not have been identified yet to be included in the ASD data.

Line 204 -- What is the reference for “1:5 male/ female ratio of idiopathic autism”? And is this stated correctly? This sentence is used following presentation of the data of 20 males with ASD in the more advanced pregnancies compared to 4 females.

Line 265 – typo “especially form the less concerning group of patients born,” there are other suspected typos throughout. (e.g. line 367 intervention. In: es, I., Levine J, editor. Neuroprotection in Autism, Schizophrenia and. . . )

Discussion: The discussion somewhat buries the lead in the statement (line 254) “However, after 33 weeks the incidence remains elevated and does not continue to decrease, following the pattern of increased risk in males, in the same gender distribution as "idiopathic" autism (33)”

Following that, it is surprisingly light on discussion of females with ASD, which is a very current topic in the autism community right now. Since the discussion acknowledges that the finding of sustained risk for males not only in prematurity but throughout gestation time is not novel, I think more discussion about the meaning of the finding for females is expected here, as that is the novel finding and the title of the article. Much is made of the findings for multiple births, which is also less than novel.

6. PLOS authors have the option to publish the peer review history of their article (what does this mean?). If published, this will include your full peer review and any attached files.

Reviewer #1: No

Reviewer #2: Yes: Michael William Gaynon MD

Reviewer #3: No

---

## [Author Response · Author response to Decision Letter 0]

24 May 2020

Here are our specific corrections and comments: 

# Academic editor

PONE-D-19-35927

Autism risk linked to prematurity is more accentuated in girls

PLOS ONE

Author response: 

Ok, Thank you.

Reviewers' comments:

Reviewer's Responses to Questions

#Comments to the Author

1. Is the manuscript technically sound, and do the data support the conclusions?

Reviewer #1: No

Reviewer #2: Yes

Reviewer #3: Yes

Author response:

We thank the reviewers, as per the comments of the reviewers, changes were made to the article. 

2. Has the statistical analysis been performed appropriately and rigorously?

Reviewer #1: No

Reviewer #2: Yes

Reviewer #3: Yes

 Author response: 

In accordance with the comments of reviewer 1, changes in the statistical analyses were made in the article

3. Have the authors made all data underlying the findings in their manuscript fully available?

Reviewer #1: No

Reviewer #2: Yes

Reviewer #3: No

Author response:

In accordance with the comments, we now allow access to all anonymous research data, hence we will need to translate the information to English. 

4. Is the manuscript presented in an intelligible fashion and written in standard English?

Reviewer #1: Yes

Reviewer #2: Yes

Reviewer #3: Yes

 Author response:

Thanks.

5. Review Comments to the Author

Reviewer #1

Reviewer #1: This study examines the association of ASD with preterm birth in 416 children from the ages of 2 to 11 years. More specifically, the authors seek to investigate the possibility of a gestational age gradient such that the risk of ASD decreases with each week of gestation from 24 to 37 weeks (from the lower boundary of extremely preterm to the upper boundary of later preterm) and how such a gradient might differ between males and females. They find that females have a significantly increased risk of ASD relative to males at lower gestational ages, whereas males maintain the same risk of ASD regardless of degree of prematurity.

Comments

1. The authors’ claim that such a gestational age gradient “has not been evidenced” in prior studies is inaccurate. At least two prior studies have demonstrated such a gradient (by week) with much larger epidemiological samples and with adjustment for a number of critical covariates and confounding variables that were not considered in the current study, presumably because the small sample size precluded their inclusion in statistical analyses. Please see Kuzniewicz et al., 2014 and Xie et al. (2017). In addition, Talmi et al. (2020) recently conducted a retrospective national cohort study of all children born in Israel from 2000-2012 using gestational age categories of extreme, very, moderate, and late preterm, showing a clear-cut gestational age gradient, as have a number of other large European population studies over the past decade, none of which are cited by the authors.

Author response:

We thank the reviewer for the constructive comments. We now addressed the aforementioned studies in our introduction. The analysis in the studies was categorical and divided prematurity in three large groups based on three levels of prematurity (uneven groups in terms of weeks- comparing two to six weeks clumped together). Other studies mentioned by the reviewer analyze birth weight and not gestational week. Our study offers risk data per each gestational week as a continuum and emphasizes the gender differences, which were not demonstrated in prior studies. 

2. The authors do not adequately address the loss to follow up of more than half of the sample (n=871) that was referred to them and the potential bias that this introduces into their results. At the very least, demographic and birth characteristics between those retained and those lost should be compared.

Author response:

Thank you, we added demographic and birth characteristics between those retained and those lost: 

"Of the 838 children identified who were born premature, 416 children were seen after the age of two years; 422 children were excluded. Of the excluded 229 (54.3%) were male, and 193 (45.7%) were female. The average birth week was 32.3 weeks (SD=2.5) and the average birth weight was 1670 grams (SD=486), which is significantly higher compared to children who were seen after the age of two years, t(834)=7.26, p<0.001, t(819)=6.65, p<0.001, respectively". 

3. Fetal growth restriction has been found to contribute, independently of gestational age, to ASD risk in a number of high-quality studies. The lack of consideration of FGR/SGA in this study is problematic, especially in that FGR may represent a distinct preterm phenotype that confers ASD risk by different mechanisms than GA.

Author response: 

Thank you for your comments, we re-analyzed the data. Effect for IUGR was not significant and the conclusions remained the same. We added:

"We also examined IUGR/SGA. Thirty seven children had IUGR/SGA, 20 were males (54%) and 17 were females (46%). Birth week ranged from week 24-37, with an average of 30.8 weeks (SD= 3.9). Birth weight ranged from 580-2200 grams, with an average of 1130 grams (SD=448).

No correlation was found between IUGR/SGA and ASD, Ҳ² (1)=0.2, p>.05., Thirty four of IUGR without ASD (92%) and 3 IUGR with ASD (8%). In order to examine whether IUGR had an effect on the results we exclude these 37 children from model. The model was significant Ҳ²(5)=17.74, p<0.01, effect for IUGR was not significant Ҳ²(1)=0, p>0.05 and the conclusions remained the same: the probability to ASD decreased for females with each birth week progress while for males it remained relatively constant".

4. The large age range is somewhat problematic in that the preterm ASD phenotype is not well characterized. ASD classifications in the youngest children, whether positive or negative, may not be stable.

Author response:

Thank you for affirming our approach, this is indeed the reason we included only the children that were in the 416 cohort, with confirmed ASD after age 2 years, in the model and multiple comparisons.

5. It is now standard in autism epidemiological research to investigate ASD with and without intellectual disability (ID), as these are likely to differ to in etiology. The small sample size of the current study may have precluded meaningful statistical analysis addressing ID, but these data could have been presented for descriptive purposes. For example, one wonders how many of the EPT/VPT girls with ASD had ID and severe ID. The authors list a large number of cognitive measures, but they do not use these to describe their sample.

Author response:

Thank you for this important comment, the requested analysis strengthened significantly our model and revealed an important finding: 

Indeed, we have followed your indication and examined the effect children with ID might have on the model, but found no contribution of this factor (ID) to the model. Moreover, children with ID were included in the group of 37 children with IUGR. Thus, by excluding them all in the model without IUGR and obtaining similar results, we confirm that no added effect is measured there. 

Since the study group is young, we also added a model including GDD comorbidity (as a precursor of ID):

"68 children (16.3%) received a diagnosis of GDD, thirteen of them have ASD (19%). Adding the GDD as an intervening factor to the model, strengthened the model slightly Ҳ²(6)=22.81, p<0.001 and the GDD is significant Ҳ²(1)=6.48, p<0.05 as well as its interaction with birth week and gender Ҳ²(1)=4.22, p<0.05. When constructing a graph based on the nominal logistic model prediction, we see that while for females trends remain similar, the proportion of ASD drops as birth week increases. For females with GDD, the same trend is kept but with a higher propensity towards ASD. For males however, trends reverse. While for males without GDD the trend is relatively constant across birth weeks, for males with GDD the proportion of males with ASD increases with birth week (See Figure 4)".

6. The authors state that all ASD diagnoses were confirmed with the ADOS, but do not report any ADOS data, for example, the Modules used – which would have given a better sense of the level of function in the sample – or the severity scores for repetitive behaviors as well as social communicative impairment. These data are of interest in the characterization of any preterm ASD sample, given that the ADOS, a gold-standard measure, has rarely been used to characterize ASD in premies.

Author response:

Thank you, the ADOS data was not included since the sample comprised of young children. For most of the participants, Module 1 was used, more recently the Toddler adaptation for Module 1. Because of the age group and comorbidities related to prematurity, we assumed that severity data will not be conclusive. Based on the remarks, we are in the process of expanding the study and examining further the clinical symptoms of ASD in our premature group in comparison to a term group, but the results are beyond the scope of this paper. 

7. The rationale for investigating the GA gradient for global development delay (GDD) and CP in addition to ASD is not clearly or compellingly explained. It is not clear if these are overlapping or mutually exclusive groupings, although the latter seems to be the case. In addition, it is not reported if any of the children classified as ASD had severe ID or severe gross motor impairment. This is problematic in that DSM-5 excludes an ASD diagnosis that is “better explained by intellectual disability (intellectual developmental disorder) or global developmental delay”, and it is not uncommon for EPT-born children to have this severity of ID. In addition, severe motor impairment (e.g., a Gross Motor Function Classification of Score = 5) could result in autism-like symptoms because of an inability to orient, direct attention, show, use gestures, etc.

Author response:

Thank you for your useful comments, we added:

"68 children (16.3%) received a diagnosis of GDD, thirteen of them have ASD (19%). Adding the GDD as an intervening factor to the model, strengthened the model slightly Ҳ²(6)=22.81, p<0.001 and the GDD is significant Ҳ²(1)=6.48, p<0.05 as well as its interaction with birth week and gender Ҳ²(1)=4.22, p<0.05. When constructing a graph based on the nominal logistic model prediction, we see that while for females trends remain similar, the proportion of ASD drops as birth week increases. For females with GDD, the same trend is kept but with a higher propensity towards ASD. For males however, trends reverse. While for males without GDD the trend is relatively constant across birth weeks, for males with GDD the proportion of males with ASD increases with birth week (See Figure 4)".

In regards to children with an additional severe motor diagnosis we included it as a formal diagnosis of cerebral palsy. It is true that the clinical symptoms of severe CP or of global developmental delay may overlap with autism symptoms, but this is the main clinical expertise of our center- to be able to differentiate between CP or GDD as a main diagnosis verses CP or GDD as a comorbidity to an ASD diagnosis. Our diagnostic team is comprised not only of neurodevelopmental doctors and psychologists, but also occupational, language and physical therapists. We assess all children by formal assessments for all developmental diagnoses enabling specific and accurate differential diagnosis. Description of our assessment of CP was published in Gabis et al., 2015.

8. The authors do not adequately account for multiple birth in their statistical analyses. Approximately, 44% of the sample and 44% of those diagnosed with ASD were multiples, with the large majorities of those twins. (Information on monozygosity was not available.) T-tests and chi-squares are used to analyze differences between singletons and multiples in mean GA and distributions by gestational age, respectively, but the results are difficult to follow and the authors do not offer a clear interpretation. It would seem more effective to have included this variable as a covariate in the logistic regression analysis. I also wondered how many of those classified with ASD were from the same birth.

Author response:

Thank you, we added:

"Out of 184 multiple pregnancies 19 had ASD (10.3%). We examined the distribution of the proportion of multiple pregnancies with ASD across birth weeks but no particular significance was found between earlier and later birth weeks, nor were we able to construct a significant model indicating twins as a significant intervening factor Ҳ² (1)=1.52, p>.05."

9. It seems necessary to consider that the present findings are unreliable, given the small sample size and the inconsistency of the findings from a number of prior studies, which are not addressed in this report. It is notable, however, that others (e.g., Schendel & Bhasin; Xie et al., 2014) have found that females are at relatively higher risk of ASD with ID vs. without ID than males, which may bear on the current findings, if ID were taken into account.

Author response:

We thank the reviewer for the important comment. Indeed, similar findings suggestive of a relationship between prematurity and autism were found in other older studies (Schendel & Bhasin, 2008), and also newer studies (Xie et al., 2017). We discussed limitations and added value of our study in the introduction as referring to the above comment. 

Reviewer #2

Reviewer #2: This manuscript explores the relationship of birth week among premature infants to Autism Spectrum Disorder (ASD), to see if the degree of prematurity might predict the likelihood that ASD might develop. 416 children born from 2011 to 2017, from an original cohort of 871 premature children, followed for at least two years were studied retrospectively, looking at birth week and gender. All children were from a single child development referral center in Israel. Nearly half were twins or triplets, some of whom may have been monozygotic. The authors acknowledge that this may have affected the risk for developing ASD. A total of 43 children were felt to have ASD, and the risk was greatest for those who were more premature. While the overall risk of developing ASD is greater for males than females, an elevated risk of ASD with lower birth week was greatest for females, who had a higher risk than males for lower birth weeks and a lower risk than males for higher birth weeks. For males, the risk was relatively stable across the range of birth weeks.

ASD is felt to be due to a combination of genetic and environmental factors, with brain insult and inflammation possibly contributing to the risk of developing it. Factors related to premature birth may contribute to the environmental components contributing to the development of ASD. Lower birth weight correlated with ASD risk in females, but not in males. A possible correlation between Cerebral Palsy (CP) and Global Developmental Delay (GDD) was also looked for. Measurements of ASD, CP and GDD by gender are shown across the range of birth weeks in the Figures 1-3. Table 1 summarizes the incidence of ASD by gender and birth week. A number of measures of cognitive function were utilized. Statistical analysis was appropriate.

This study shows that prematurity is a risk factor for the development of ASD, and provides additional information regarding the relative risk attributable to birth week and gender. It is beyond the scope of this study to look at specific factors that might contribute to the risk of ASD in premature children, such as anoxic brain damage, intraventricular hemorrhage, presence or absence of Retinopathy of Prematurity (ROP), and treatment of Type 1 ROP with anti-VEGF injections. What can be said is that earlier intervention for those developing ASD may contribute to better outcomes, and recognition of the increased risk of ASD in premature children is therefore important. It is an interesting study.

Author response:

Thank you

Reviewer #3

Reviewer #3: This is an interesting study with a retrospective look at prematurity and gender risk for ASD. There are several things that can strengthen it to make it more meaningful to autism researchers, clinician, and parents.

Abstract: Children were followed for only “at least” two years, so the statement about girls should may need to be tempered to say that this relationship is evident only for girls whose symptoms are sufficiently apparent to be diagnosed by the age of 24 months, if that was the case (no info is given on when they were diagnosed). Girls with higher cognitive and language abilities are often not diagnosed until much later.

Author response:

We agree, for this reason we limited the group to two years and above. Some of the girls were indeed diagnosed later, and as such included in the analysis. However we did not find a significant age difference in the age of diagnosis. 

Introduction: When the manuscript is revised, the CDC prevalence data needs to be updated to 1:54. “significant risk of developmental risk” may need to be revised – is developmental delay the intended meaning? 

Author response:

Thank you for your attention, corrected. 

Please clarify when the children were evaluated. It is stated that you only included children seen in the clinic up to the age of 2 to prevent loss to follow up, but several of the cognitive measures listed are for older children only. At what ages where the cognitive evaluations completed? It is apparent from the age range (2-14.2 years) that cognitive testing may have occurred at any point. This is a retrospective study, so presumably cognitive testing was done for a reason unrelated to the study, which should be addressed at least briefly to determine possibility of bias in the sample. For example, if the cognitive testing was performed as a matter of routine follow up for a premature child, that is less biased than if they were only performed when disability was suspected, introducing at least some selection bias in the sample. This issue is also related to the comment above about females across the spectrum who are typically not diagnosed until later in life.

Author response:

Thank you for your remark. The cognitive evaluation is performed initially during the initial diagnostic process and, according to the DSM-5 and MOH guidelines, is mandatory when a diagnosis of ASD made in order to establish that ID is not the main diagnosis (may accompany ASD). As such, all cognitive assessments were performed at the initial evaluation of autism and the instruments used are Mullen (up to age three) and IQ tests, such as WPPSI-III (until school age) and WISC-R (school age). 

How old were the children when they were diagnosed? That is an important proxy for interpretation of the severity of the ASD.

Author response:

Thank you for your remark, we added age of diagnosis:

"The age of initial ASD diagnosis was in the range of 1.1-8.5 years, with mean age of 3.5 years (SD=1.8)". 

Also, it would be helpful to know co-morbid diagnoses. Co-morbidity data would also aid in the interpretation of the findings. Are the females diagnosed with ASD in the lower gestational ages also affected by multiple disabilities or is ASD the primary risk? All of this information is more helpful than a blanket statement about risk and females as if they are all equivalent in terms of ASD. The current data are helpful, but MORE data from the records would significantly improve the interpretation and application of the study. Since Cognitive testing was completed, that is one example of data that aids in interpretation. Females with Intellectual Disability (ID) present very differently than females without ID, and may or may not have been identified yet to be included in the ASD data.

Author response:

Thank you, we examined the effect ID children might have on the model and each pair comparisons for significance but found no contribution of this factor (ID) to the model. Moreover, children with ID were all included in the group of 37 children with IUGR. Thus, by excluding them all in the model without IUGR children and obtaining similar results we confirm that no added effect is measured there. We also added a model with GDD:

"68 children (16.3%) received a diagnosis of GDD, thirteen of them have ASD (19%). Adding the GDD as an intervening factor to the model strengthened the model slightly Ҳ²(6)=22.81, p<0.001 and the GDD is significant Ҳ²(1)=6.48, p<0.05 as well as its interaction with birth week and gender Ҳ²(1)=4.22, p<0.05. When constructing a graph based on the nominal logistic model prediction, we see that while for females trends remain similar; the proportion of ASD drops as birth week increases, for females with GDD the same trend is kept but with a higher propensity to ASD. For males however, trends reverse; while for non GDD males the trend is relatively constant across birth weeks, for males with GDD the proportion of ASD males increases with birth week (See Figure 4)".

Line 204 -- What is the reference for “1:5 male/ female ratio of idiopathic autism”? And is this stated correctly? This sentence is used following presentation of the data of 20 males with ASD in the more advanced pregnancies compared to 4 females.

Author response:

Thank you, we added reference: 

Gabis L V., Pomeroy J. An etiologic classification of autism spectrum disorders. Isr Med Assoc J. 2014;16(5):295–8.

Line 265 – typo “especially form the less concerning group of patients born,” there are other suspected typos throughout. (e.g. line 367 intervention. In: es, I., Levine J, editor. Neuroprotection in Autism, Schizophrenia and. . . )

Author response:

We thank the reviewer for turning our attention to the typos. Based on the reviewer’s note we made the change to line 273, 397, and throughout the article.

Discussion: The discussion somewhat buries the lead in the statement (line 254) “However, after 33 weeks the incidence remains elevated and does not continue to decrease, following the pattern of increased risk in males, in the same gender distribution as "idiopathic" autism (33)”

Following that, it is surprisingly light on discussion of females with ASD, which is a very current topic in the autism community right now. Since the discussion acknowledges that the finding of sustained risk for males not only in prematurity but throughout gestation time is not novel, I think more discussion about the meaning of the finding for females is expected here, as that is the novel finding and the title of the article. Much is made of the findings for multiple births, which is also less than novel.

Author response:

Thank you very much for the opportunity to include our insights on gender differential vulnerability in the developing brain. We added a paragraph and new references in the discussion. 

6. PLOS authors have the option to publish the peer review history of their article (what does this mean?). If published, this will include your full peer review and any attached files.

Author response: OK.

---

## [Decision Letter · Decision Letter 1]

9 Jul 2020

PONE-D-19-35927R1

Autism risk linked to prematurity is more accentuated in girls

PLOS ONE

Dear Dr. Gabis,

Thank you for submitting your revised manuscript to PLOS ONE. After careful consideration, there are still a few minor issues that will need to be addressed. In particular, one of the reviewers made the point that attrition could have affected your estimates. Please address this statistically or by noting in the text how this may impact your results and interpretations. The reviewer also noted a typo. Please proof your manuscript carefully as PLOS ONE will not copy edit your manuscript for typographical errors. 

We look forward to receiving your revised manuscript.

Kind regards,

Eric J. Moody, Ph.D.

Academic Editor

PLOS ONE

Reviewers' comments:

Reviewer's Responses to Questions

**Comments to the Author**

1. If the authors have adequately addressed your comments raised in a previous round of review and you feel that this manuscript is now acceptable for publication, you may indicate that here to bypass the “Comments to the Author” section, enter your conflict of interest statement in the “Confidential to Editor” section, and submit your "Accept" recommendation.

Reviewer #2: All comments have been addressed

Reviewer #4: (No Response)

2. Is the manuscript technically sound, and do the data support the conclusions?

Reviewer #2: Yes

Reviewer #4: Yes

3. Has the statistical analysis been performed appropriately and rigorously? 

Reviewer #2: Yes

Reviewer #4: Yes

4. Have the authors made all data underlying the findings in their manuscript fully available?

Reviewer #2: Yes

Reviewer #4: No

5. Is the manuscript presented in an intelligible fashion and written in standard English?

Reviewer #2: Yes

Reviewer #4: Yes

6. Review Comments to the Author

Reviewer #2: I believe that the questons posed by the three reviewers have been adequately answered. I have nothing further to add, apart from my initial comments.

Reviewer #4: line 287: the prevalence of ASD in the study sample could have been inflated by the high attrition rate (the infants without ASD are more likely to not coming back to the clinic.)

line 419: typo

7. PLOS authors have the option to publish the peer review history of their article (what does this mean?). If published, this will include your full peer review and any attached files.

Reviewer #2: No

Reviewer #4: No

---

## [Author Response · Author response to Decision Letter 1]

16 Jul 2020

# Academic editor

PONE-D-19-35927

Autism risk linked to prematurity is more accentuated in girls

PLOS ONE

6. Review Comments to the Author

#Comments to the Author

Reviewer #2: I believe that the questions posed by the three reviewers have been adequately answered. I have nothing further to add, apart from my initial comments.

Author response: Thank you. 

Reviewer #4: line 287: the prevalence of ASD in the study sample could have been inflated by the high attrition rate (the infants without ASD are more likely to not coming back to the clinic.)

Author response: 

Thank you for this very important remark and we will try to address it in different ways: 

- In regards to developmental follow up of non –ASD children, loss to follow up after the age of two years may be suspected for premature children with typical development, which may influence the rate of disability in the cohort. However, this may influence incidence of all developmental issues and not only ASD, since a high percentage of children born prematurely may have global developmental delay or delays in a single area of development.

- This matter of declining numbers in the control cohort needs to be addressed, and is in the first steps of data exploration, as explained in the abstract in rows 29-30. In order to analyze this possible bias we recalculated the incidence for the whole cohort of 817 children followed from birth, including infants that did not reach two years. The same model and multiple comparisons were implemented and still showed similar significances as in the subset cohort of 416 children with a validated diagnosis of positive/ negative ASD. Within the 391 that were removed from the main analysis cohort of 416 children, there were 389 children with no ASD diagnosis, and 2 females born week 27 and 36 (*10 exceptions excluded) with ASD. With this additional analysis, the principal conclusion was even strengthened- that the increased incidence with lower birth week is driven mainly by girls. So to answer the reviewer’s comment, an increase in controls does not seem to affect the conclusion and vice versa.

- 

line 419: typo

Author response: Corrected, thank you.

---

## [Editor Report · Decision Letter 2]

20 Jul 2020

Autism risk linked to prematurity is more accentuated in girls

PONE-D-19-35927R2

Dear Dr. Gabis,

We’re pleased to inform you that your manuscript has been judged scientifically suitable for publication and will be formally accepted for publication once it meets all outstanding technical requirements.

Kind regards,

Eric J. Moody, Ph.D.

Academic Editor

PLOS ONE
---

## [Editor Report · Acceptance letter]

11 Aug 2020

PONE-D-19-35927R2 

Autism risk linked to prematurity is more accentuated in girls 

Dear Dr. Gabis:

I'm pleased to inform you that your manuscript has been deemed suitable for publication in PLOS ONE. Congratulations! Your manuscript is now with our production department. 

Kind regards, 

on behalf of

Dr. Eric J. Moody 

Academic Editor

PLOS ONE